# Chiari 1 Malformation and Epilepsy in Children: A Missing Relationship

**DOI:** 10.3390/jcm11206182

**Published:** 2022-10-20

**Authors:** Luca Massimi, Davide Palombi, Ilaria Contaldo, Chara Veredice, Daniela Rosaria Pia Chieffo, Rosalinda Calandrelli, Gianpiero Tamburrini, Domenica Immacolata Battaglia

**Affiliations:** 1Pediatric Neurosurgery, Fondazione Policlinico Universitario A. Gemelli IRCCS, 00168 Rome, Italy; 2Department of Neuroscience, Università Cattolica del Sacro Cuore, 00168 Rome, Italy; 3Pediatric Neurology, Fondazione Policlinico Universitario A. Gemelli IRCCS, 00168 Rome, Italy; 4Clinical Psychology, Fondazione Policlinico Universitario A. Gemelli IRCCS, 00168 Rome, Italy; 5Institute of Radiology, Fondazione Policlinico Universitario A. Gemelli IRCCS, 00168 Rome, Italy

**Keywords:** Chiari 1 malformation, syringomyelia, epilepsy, seizures, posterior fossa, EEG, cerebellum

## Abstract

*Purpose*: Once believed a result of pathophysiological correlations, the association between Chiari 1 malformation (CM1) and epilepsy has since been considered as a coincidence, due to missing etiologic or clinical matching points. At present, the problem is being newly debated because of the increasing number of CM1 diagnoses, often among children with seizures. No specific studies on this topic are available yet. The present study aimed at updating the information on this topic by reporting on a series of children specifically enrolled and retrospectively analyzed for this purpose. *Methods*: All children admitted between January 2015 and June 2020 for epilepsy and CM1 were considered (Group 1). They were compared with children admitted in the same period for symptoms/signs related to CM1 and/or syringomyelia (Group 2). Syndromic patients were excluded, as well as those with tumoral or other overt intracranial lesions. All patients received a complete preoperative work-up, including MRI and EEG. Symptomatic children with CM1/syringomyelia were operated on. The pertinent literature was reviewed. *Results*: Group 1 was composed of 29 children (mean age: 6.2 years) showing CM1 and epilepsy with several types of seizures. A share of 27% had CM1-related symptoms and syringomyelia. The mean tonsillar ectopia was 7.5 mm. Surgery was performed in 31% of cases. Overall, 62% of children are currently seizure-free (including 5/9 children who were operated on). Tonsillar herniation and syringomyelia regressed in 4/9 cases and 4/8 cases, improved in 4/9 cases and 3/8 cases, and remained stable in 1/9 and 1/8 cases, respectively. CM1 signs/symptoms regressed completely in 6/8 cases and improved or remained stable in one case in each of the two remaining patients. Group 2 consisted of 77 children (mean age: 8.9 years) showing symptoms of CM1 (75%) and/or syringomyelia (39%). The mean tonsillar ectopia was 11.8 mm. Non-specific EEG anomalies were detected in 13 children (17%). Surgery was performed in 76.5% of cases (18 children were not operated on because of oligosymptomatic). Preoperative symptoms regressed in 26%, improved in 50%, remained stable 22%, and worsened in 2%; CM1 radiologically regressed in 39%, improved in 37%, remained unchanged in 22%, and worsened in 2%; and syringomyelia/hydromyelia regressed in 61%, improved in 30%, and was stable in 9%. No statistically significant differences between the two groups were detected regarding the M/F ratio, presence of syringomyelia/hydromyelia, or CM1/syringomyelia outcome; moreover, no correlation occurred between seizure-free condition and PF decompression in Group 1, or between disappearance of EEG anomalies and PF decompression in Group 2. A significant difference between the two groups was noticed regarding the mean age at admission (*p* = 0.003), amount of tonsillar herniation (*p* < 0.00001), and PF decompression (*p* = 0.0001). *Conclusions*: These findings do not support clinical correlations between CM1 and epilepsy. Their course depends on surgery and antiepileptic drugs, respectively. The analysis of the literature does not provide evidence of a relationship between seizures and cerebellar anomalies such as CM1. Rather than being linked to a syndrome that could explain such an association, the connection between the two now has to be considered to be random.

## 1. Introduction

The association between Chiari type I malformation (CM1) and epilepsy raises relevant interest among clinicians and researchers. The first reason for such an interest is related to the growing number of new diagnoses of CM1, both in children and adult [1,2]. The increasing numbers of patients with diagnosed CM1 (and syringomyelia) require increasing efforts for their correct classification and treatment, a need that is exacerbated by the still-missing guidelines about CM1 management [3]. One of the results of the greater “attention” paid to CM1 patients is their division into different subtypes [4], of which the CM1/epilepsy groups are potentially one. On the other hand, epilepsy is already known to be a common problem, with a 0.5–1% prevalence in the general population [5] and 8% mean prevalence in hospitalized patients worldwide [6]. Therefore, its impact on the population’s health is significant.

On these grounds, the association between the previous two conditions is regarded with concern because of the known impact of epilepsy in the clinical practice and the potential role of CM1 in further increasing that impact. A peculiar issue, which has been addressed in the past, regards the possible etiological and clinical relationships between the two conditions. No conclusions have been reached so far about whether this association may be considered to be random. The present study aims to update the information on this topic by reporting on a series of children specifically enrolled and analyzed for this purpose and by reviewing the pertinent literature.

## 2. Materials and Methods

All children consecutively admitted between January 2015 and June 2020 for epilepsy and harboring CM1 malformation were considered for the present study. They were compared with children admitted in the same period for symptoms/signs related to CM1 and/or syringomyelia. The aforementioned time period was considered to include patients with homogeneous preoperative and postoperative evaluation, and to have a minimum follow-up of 2 years.

Epilepsy and seizures were classified according to ILAE 2017 classification [7]. Epilepsy was managed by antiepileptic drugs according to the patients’ characteristics. CM1 was radiologically defined as tonsillar caudal ectopia ≥ 5 mm, associated with peg-like appearance of the tonsils or other signs of posterior fossa (PF) overcrowding (e.g., effacement of CSF spaces) [8]. Surgery was performed in clearly symptomatic patients and/or with large or progressing syringomyelia [3]. PF decompression was performed to manage CM1, by suboccipital craniectomy and C1 laminectomy, with/without dural opening (and tonsil shrinkage) based on the patient’s characteristics. Endoscopic third ventriculostomy was used to manage associated hydrocephalus.

All children received preoperative and postoperative neurological examination, neuropsychological tests, brain (with perfusion sequences) and whole spinal cord MRI, and surface EEG. In selected cases, accessory investigations, such as motor and/or somatosensory potentials and polysomnography, were performed (their results are not discussed here since their impact was not relevant). Syndromic patients were excluded, as well as those with tumoral or other overt intracranial lesions (e.g., arachnoid cysts), incomplete preoperative/postoperative work-up, or details lost at follow-up.

Patients with epilepsy and CM1 were mainly referred to the Pediatric Neurology department of our institution, while those with CM1 were referred to the Pediatric Neurosurgery department. The first subset of patients was included in Group 1 of the present study, while those with CIM belonged to Group 2. The mean follow-up was 40.68 months (range: 24–66 months).

The following variables were considered for the comparison between the two groups: age, sex, presence of EEG anomalies, cerebellar tonsils herniation, presence of hydromyelia/syringomyelia, PF surgical decompression, postoperative regression/improvement of CM1 symptoms, and radiological picture. The statistical analysis was realized the through χ^2^ test and 2-tailed *t*-test. A *p* value < 0.05 was considered as statistically significant.

## 3. Results

### 3.1. Patients’ Characteristics 

Overall, 106 children were enrolled for the present analysis. All of them were alive at current follow-up.

Group 1 was composed of 29 children (14 boys, 15 girls) with a 6.2-year mean age at admission (range: 11 months–15 years). All these patients were admitted and studied because of epileptic seizures and all of them were found to have CM1. These children represent 12% of all pediatric patients admitted to our institution for epilepsy in the considered period and 18% of all CM1 cases (both symptomatic and incidentally diagnosed) observed in the same period (157 cases). Regarding seizures, 4 children (13%) presented absences (A), 8 (27.5%) focal motor seizures (FM), 6 (20.5%) focal non-motor seizures (FNM), 4 (13%) focal to generalized tonic-clonic seizures (FGTC), 2 (10%) generalized tonic-clonic seizures (GTC) (Figure 1), 4 (13%) generalized tonic seizures (GT), and 1 (3%) myoclonic seizure (M). All children underwent antiepileptic drug treatment. Eight out of 29 children had signs/symptoms related to CM1 (27%). The mean tonsillar ectopia was 7.5 mm (range: 5–13 mm) (Figure 2). Eight patients (27%) harbored syringomyelia (5 cases) or hydromyelia (3 cases), and 2 cases were associated with hydrocephalus. No other relevant anomalies or microlesions were detected on MRI. The details of Group 1 patients are reported in Table 1.

Group 2 consisted of 77 children (43 boys, 34 girls) with an 8.9-year mean age at admission (range: 3–18 years). All these children were admitted because of CM1 with/without syringomyelia. All of them were symptomatic, but signs/symptoms clearly related to CM1/syringomyelia were present in 75% of cases (58 children) (Figure 3). Occipital cough and headache were the main symptoms, followed by dizziness, nystagmus, ataxia, paresthesia, and dysphagia. The mean tonsillar ectopia was 11.8 mm (range: 5–28 mm). Thirty patients (39%) harbored syringomyelia (21 cases) or hydromyelia (9 cases), and 3 cases were associated with ventriculomegaly. No other relevant anomalies or microlesions were detected on MRI. EEG anomalies were detected in 13 children (17%), being represented by slow waves in 6 cases (Figure 4), theta activity in 3 cases, sporadic spikes in 2 cases, and sharp waves, spikes-waves, and poli-spikes in the remaining 2 cases. None of these 13 children showed epileptic seizures nor received antiepileptic drugs. The details on Group 2 patients are reported on Table 2.

### 3.2. Treatment and Outcome

In Group 1, surgery (for CM1 and/or ventriculomegaly) was performed in 9 cases (31%): PF decompression was realized in 8/9 cases (in one patient following ETV), while ETV alone was performed in the remaining patient. Five of the 9 children operated on are seizure-free. Overall, 62% of children are currently seizure-free (18 cases). On neuroimaging, tonsillar herniation and syringomyelia/hydromyelia regressed at follow-up in 4/9 cases and 4/8 cases, improved in 4/9 cases (including the patient who received only ETV) and 3/8 cases, and remained stable in 1/9 and 1/8 cases, respectively. CM1 signs/symptoms regressed completely in 6/8 cases (one patient was asymptomatic but operated on for large syringomyelia, one underwent ETV alone) and improved or remained stable in one case each of the two remaining patients. Among the 21 non-operated-on cases, the PF radiological picture remained substantially unchanged, except for 2 cases where the tonsillar herniation spontaneously improved (Table 3). 

In Group 2, surgery was performed in 59 cases (76.5%): PF decompression was realized in 54/59 cases because of symptomatic CM1 and/or large or progressing syrinx, while the remaining 5 patients received ETV for hydrocephalus (3 cases) and invasive recording of intracranial pressure (ICP), which ruled out raised ICP. The remaining 18 children were not operated on because they were oligosymptomatic. At follow-up, 15 out of these 18 children had their radiological picture unchanged, while a spontaneous improvement in CM1 was detected in the remaining 3. All 3 patients undergoing ETV had normalization of hydrocephalus and improvement in their CM1/syringomyelia-related clinical and radiological picture. Finally, among patients operated on for CM1/syringomyelia: preoperative symptoms regressed in 14 cases (26%), improved in 27 (50%), remained stable in 12 (22%), and worsened in one (2%); on neuroimaging, CM1 radiologically regressed in 21/54 children (39%), improved in 20/54 (37%), remained unchanged in 12/54 (22%), and worsened in 1/54 (2%); and in syringomyelia/hydromyelia, regressed in 14/23 children (61%), improved in 7/23 (30%), and was stable in 2/23 (9%) (Table 4).

### 3.3. Comparison between the Two Groups

No statistically significant differences between Group 1 and Group 2 were detected regarding M/F ratio (0.93 vs. 1.24, respectively) and presence of syringomyelia/hydromyelia (27% vs. 39%). Moreover, no relevant differences were found in the CM1/syringomyelia outcome, namely, about regression/improvement of preoperative signs/symptoms (87.5% vs. 76%), radiological regression/improvement of tonsillar ectopia (88% vs. 76%), and syringomyelia/hydromyelia (87.5% vs. 91%). In addition, no correlation was found between seizure-free condition and PF decompression in Group 1, or between disappearance of EEG anomalies and PF decompression in Group 2, although the significance of these data is not reliable because the number of observations is too small (29 and 13 cases, respectively) (Table 5).

On the other hand, a significant difference between the two groups was noticed in terms of mean age at admission (6.2 vs. 8.9 years, *p* = 0.003), amount of tonsillar herniation (7.5 vs. 11.8 mm, *p* < 0.00001), and PF decompression (27.5% vs. 70%, *p* = 0.0001). A final, obvious difference was found in the occurrence of EEG anomalies (100% vs. 17%, *p* < 0.00001) (Table 5).

## 4. Discussion

Several studies, conducted mainly at the end of the last century and the beginning of the current one, raised the possible physiopathological association between CM1 and epilepsy. However, although this was clinically hypothesized, it has not been demonstrated yet [9,10,11,12,13]. The aim of the present study was to provide an update on possible clinical correlations between these two conditions, taking into account the evolution in the classification of both epilepsy and CM1. Indeed, the present study, if compared with the previous ones, included the new classification of epilepsy [7] and the new concepts relating to the CIM definition, which is no longer considered a mere tonsillar descent alone [3]. Moreover, the current study is based on a pediatric population admitted at a single institution and investigated homogeneously.

A first issue resulting from our study concerns the epidemiological implications of the CM1 and epilepsy association. Historical, large surgical or radiological series showed a prevalence of epilepsy among CM1 patients ranging from 3.8% to 10% [14,15,16]. In the present experience, 18% of 157 CM1 patients followed in the considered 5-year period (77 children symptomatic for CM1, which are here included, and 80 asymptomatic) were found to present epileptic seizures. The main explanation for this higher rate may be selection bias, because our institution is a tertiary referral center for both CM1 and epilepsy; therefore, the number of symptomatic subjects with both conditions is concentrated. However, in more recent studies, such as that of Marianayagam and coworkers, similar rates are reported (13.8%) [17]. This could depend on the increasing number of CM1 diagnoses and on the more extensive diagnostic work-up that epileptic (and CM1) patients currently undergo. Such an epidemiological association is also well-known in Cavalier King Charles Spaniels dogs (CM1 77%, epilepsy 28%) where, however, no relevant correlations between the conditions have been found [18]. Similar considerations can be made about the prevalence of CM1 in epileptic subjects, which was as high as 12% in the present series but about 5% in historical series [19]. On the other hand, the occurrence of ventriculomegaly or hydrocephalus does not seem to be specific to our series, since it was sporadic and did not affect the occurrence of seizures in both groups. The treatment of this condition was based on ETV because it has been largely demonstrated that CM1-associated hydrocephalus is obstructive [20]. ETV was used as first-line treatment in the management of such an association, and also achieved good results for the improvement of CM1 or syringomyelia, as demonstrated elsewhere [21]. ETV results also a “protective” factor against epilepsy in hydrocephalic children if compared to a ventriculo-peritoneal shunt [22,23].

A second issue is related to the characteristics of patients with epilepsy and CM1. According to the literature, benign focal epileptiform discharges (mainly in the frontal and in the centrotemporal regions) and complex partial seizures are the most common types of seizures, followed by partial seizures with/without generalization, generalized tonic seizures, and myoclonic seizures [10,11,12,13,15,19]. Generally, a good epileptic outcome is reported. The present series partially confirms the trend of the types of epilepsy, with a prevalence of focal seizures (both FM and FNM, 48%) over generalized seizures (A, GT, GTC, M, 39%) and with a 13% rate of FGCT. Instead, the good epileptic outcome is totally confirmed, 62% of patients being seizure-free and 31% showing a >90% reduction in their seizures (see Table 3). The types of seizures and their outcome do not allow the formulation of relevant hypotheses but only the acknowledgement of the differences that exist among the different series. The type of seizures, in particular, depends on the characteristics of enrolled patients. Our series offers a reliable panorama of epilepsy in CM1 children because it is composed only of consecutive patients without possibly confounding factors (such as tumors or other lesions, syndromes etc.). On the other hand, the usually good epileptic outcome could be explained by the low rate of children with encephalopathy or severe brain damage, which are sporadically associated with CM1 (or which, sometimes, may “obscure” the presence of CM1, so that it could not be adequately reported in this subset of patients). This is quite different from the observations in general series of epileptic patients, where some factors, such as young age, generalized seizures, or use of a combination of drugs, can predict the remission of seizures [24].

The clinical characteristics of our series confirm another typical aspect of epileptic children with CM1, which is the high rate of associated developmental delay. In Group 1, indeed, cognitive and/or behavioral problems affected about 45% of cases (see Table 1) and, in the literature, this rate is even higher, reaching about 100% in some series [10,11,12,13,15,19]. This association is the most important clinical factor at the base of the theories postulating the presence of a common denominator between epilepsy and CM1 that can explain their possible intercorrelation. The brain (and cerebellar) microdysgenesis was hypothesized, in old studies, as a common denominator of a possible maldevelopmental process explaining the co-existence of CM1 and epilepsy. Grosso et al., for example, proposed the hypothesis that the association among CM1 and mental retardation, speech delay, and epilepsy (that they found in nine children) is not occasional but could be the expression of a specific developmental disorder [12]. However, such an association was purely clinical, and no genetic validation was provided. Moreover, not all the reported patients showed epilepsy and/or EEG anomalies. No significant correlation between tonsillar descent and epilepsy was found. Similarly, Brill et al. described a series of 11 CM1 children with epilepsy, developmental delay in motor or language function, and autistic features in some of them [10]. The authors formulated the hypothesis of a “subtle cerebral dysgenesis” that could give reason of a non-incidental occurrence of CM1 in this subset of patients. A comparison with Chiari II malformation (CM2) and its associated, dysgenetic brain findings was used to support this theory. However, CM2 is a completely different entity and, once again, the number of studied cases was too small, and no controls were analyzed. The hypothesis of a microdysgenesis was also promoted by Elia et al. on a similar series of seven children based on the absence of manifest dysgenetic findings [11]. An attempt at demonstrating the presence of a microdysgenesis was realized by Iannetti and coworkers by SPECT in four children with CM1 and epilepsy [13]. The authors found hypoperfusion brain areas in all cases, and cerebellar hypoperfusion areas in two of them. Unfortunately, such an interesting study was not replicated in a larger number of cases. Actually, no conclusions can be deduced from such a small number of cases, especially considering that the presence of cerebellar anomalies in only two cases could represent an incidental association. Moreover, one could speculate that the cerebellar anomalies could be the result of an impairment on the connectivity between the brain cortex and cerebellum, as recently demonstrated [25]. However, in these instances, epilepsy is thought to cause cerebellar anomalies, while it would not result from them or, considering the topic of the present article, from CM1.

On these grounds, the aforementioned studies did not prove a relationship between CM1 and epilepsy even though the occurrence of a syndrome, which could support this hypothesis, cannot be ruled out. Actually, the presence of a syndrome might justify the co-existence of CM1 and epilepsy because the syndrome itself can encompass a spectrum of neurological disorders including these two entities. In syndromes, seizures usually result from the syndrome-associated lesions, as happens, for example, in neurofibromatosis-1 [26,27], or from the brain damages included in the syndrome, as occurs, for example, in fetal alcohol spectrum disorders [28]. An association between CM1 and epileptogenic anomalies (e. g. focal cortical heterotopia, cortical dysplasia, holoprosencephaly) is proved outside syndromes [29]. However, although it cannot be excluded, a shared genetic and embryologic maldevelopment pattern between these malformations and CM1 is not demonstrated yet [30]. KCTD7 deficiency is an example of a shared genetic pathway involving the brain (epilepsy) and cerebellum (degeneration of Purkinje cells) [31] and could be used as a possible model for research on this topic. To date, a syndrome with a “pure” association between CM1 and epilepsy has not been identified in spite of the recent description of new syndromes, such as the PTEN hamartoma tumor or the SETD2 neurodevelopmental one, which include CM1 and epilepsy [32,33]. Once again, it is worth noting that, also in these syndromes, seizures are likely to result from associated brain anomalies (brain tumors, vascular malformations, ventriculomegaly, macrocephaly) rather from cerebellar abnormalities or CM1. To date, there is no evidence that CM1 or syringomyelia can affect the electrical activity of the brain to cause seizures.

A further issue for discussion is the potential epileptogenicity of the cerebellum and, as a consequence, of CM1. Although it is a retrospective analysis, the present study did not highlight the cerebellar anomalies on neuroimaging (even in children undergoing perfusion sequences or 3T MRI) of both groups of patients, apart those related to CM1. Buoni et al. reported on three children with asymptomatic CM1 and EEG anomalies whose EEG abnormalities regressed after PF decompression [9]. The authors theorized that the CM1-dependent CSF flow reduction (found during surgery) could interfere with the brain electrical activity (rather than possible cerebellar anomalies), which is therefore normalized after the restoration of a proper CSF flow. Several limits (some of them acknowledged by the authors themselves) affect this study and its intriguing hypothesis: (1) the patient sample is too small; (2) EEG could have improved spontaneously over the time; (3) one out of three children was epileptic, thus showing different reasons than CM1 to show seizures; in this case, however, IRDA, spikes, and spikes waves disappeared after surgery, but it is not reported if the patient received antiepileptic drugs; (4) the interpretation of EEGs may have not been correct; and (5) a microdysgenesis was not demonstrated (although it cannot be denied). Several previous studies supported the hypothesis of the role of the cerebellum in generating epilepsy, often based on animal experimental models [10,28,29,30]. Nevertheless, experimental and clinical studies failed in demonstrating an effect of cerebellar stimulation in epilepsy [34,35]. The cerebellum could be the target for a stimulation in patients with epilepsy-induced degeneration of cortical-cerebellar pathways, which could improve the course of epilepsy [25]. Of course, this is something different from supporting a systematic role of cerebellum in epileptogenesis, which was sporadically demonstrated in anecdotal cases with cerebellar masses [36,37]. Recent volumetric studies comparing epileptic subjects and healthy controls, indeed, showed volumetric anomalies in the brain, lateral ventricles, and putamen, but not in the cerebellum [38]. On these grounds, the role of the cerebellum and, in particular, CM1, in epilepsy remains undemonstrated. This conclusion is further supported by some confusing symptoms mimicking seizures in CM1 patients (and, thus, potentially reducing the prevalence of “true” seizures in CM1), such as cerebellar fits, sleep disordered-breathing, paroxysmal dyskinesias, and syncope [15,39,40].

In summary, the clinical and experimental experiences from the literature lead to the conclusion that the association between epilepsy and CM1 is random [15,17]. The present study adds some clinical findings supporting the hypothesis of such a casual association. They can be summarized as follows: (1) epileptic patients (Group 1) show well-defined EEG patterns; in contrast, these are missing in CM1 patients, where EEG anomalies are not numerically relevant nor electrically specific but sporadic (Group 2); (2) no MRI anomalies suggesting microdysgenesis were found (in both Groups); (3) children with epilepsy and CM1 represent a substantially different population of patients compared with those with CM1 alone, as shown by some common, crucial findings, such as age at admission, amount of tonsillar descent, and need of PF decompression; and (4) the course and the outcome of one condition is not affected by the other, with the outcome of epilepsy depending on the drug treatment, while that of CM1/syringomyelia depending on surgery. These considerations were shared by most of the international experts attending the 2019 Consensus Conference on CM1 and Syringomyelia as part of their daily clinical experience [3]. Indeed, 94.1% of these experts concluded that, “in children with epilepsy and CM1, surgical treatment of CM1 does not improve the seizure disorder” because of the random association between the two conditions.

### Limitations of the Study

A first limit of the present study is the retrospective analysis of a relatively small number of patients. Indeed, although it is the first study specifically addressing the problem of CM1 and epilepsy in the modern era, the small sample of patients does not allow the results to be considered as definitive. Moreover, as mentioned in the text, the statistical analysis was not reliable, in some instances, for the same reason. A multi-centric and prospective study would allow this limit to overcome to be overcome.

A second limit is the use of “rough” parameters for the quantification of CM1. We are planning to analyze new cases with multi-parameter and 3D measurements of the posterior fossa.

## Figures and Tables

**Figure 1 jcm-11-06182-f001:**
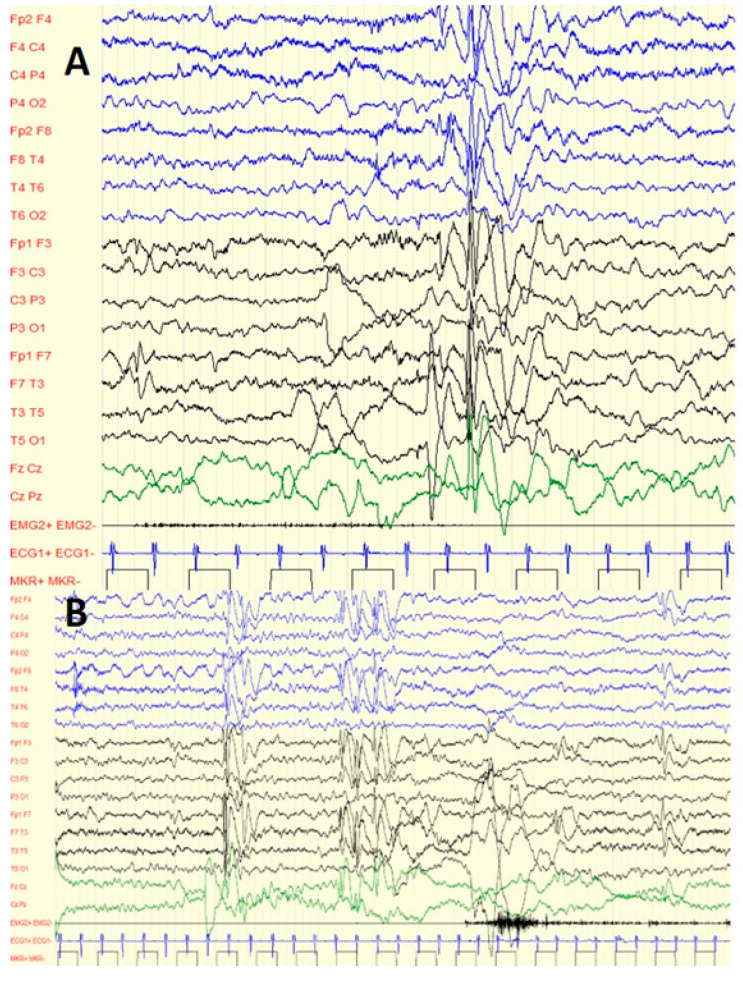
Six-year-old boy with C1M and GTC: (**A**) interictal diffuse spike waves during sleep; (**B**) interictal focal spike waves (fronto-central regions) during sleep.

**Figure 2 jcm-11-06182-f002:**
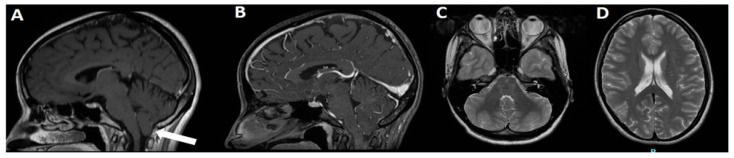
3T MRI of the same case in Figure 1. On sagittal view, C1M is evident, with moderate hypoplasia of PF and 8 mm ectopia of cerebellar tonsils (**A**, arrow). No relevant findings are visible on sagittal view after gadolinium administration (**B**) or on T2 axial views (**C**,**D**).

**Figure 3 jcm-11-06182-f003:**
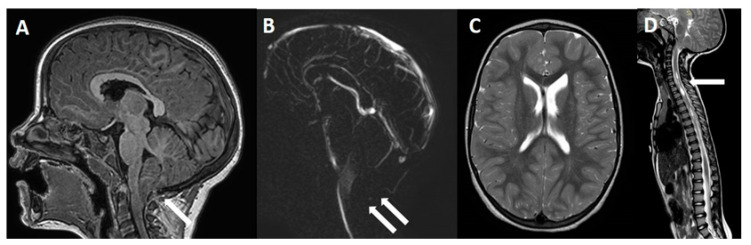
Four-year-old girl with symptomatic C1M. (**A**) MRI, sagittal view, showing a PF moderate hypoplasia with peg-like appearance of the cerebellar tonsils that present a 15 mm downward herniation below the McRae line (arrow). Note the disappearance of the CSF flow through the posterior subarachnoid spaces on cine-MRI (**B**, double arrows). No relevant findings are found on brain axial view (**C**), while a cervico-thoracic syringomyelia is evident on spinal cord MRI (**D**, arrow).

**Figure 4 jcm-11-06182-f004:**
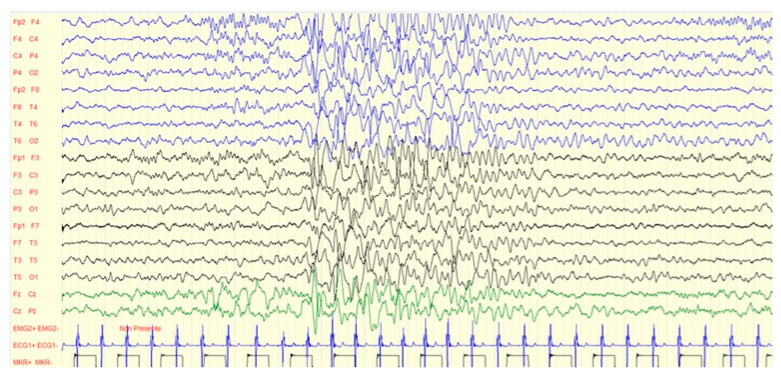
EEG anomalies (diffuse slow waves) of the same case in Figure 3.

**Table 1 jcm-11-06182-t001:** Clinical characteristics of Group 1.

Patients (No and Sex)	Age (Years)	Seizures	Interictal EEG	Developmental Problems	Tonsils Ectopia (mm)	Spinal Cord and Other Findings	C1M/Syringomyelia Symptoms
1, F	7	FM	bilateral S and SW	No	9	Normal	no
2, F	1	GT	bouffées of slow W with S (parieto-occipital region bilaterally)	No	13	cervico-thoracic syringomyelia	cough, headache, ataxia
3, F	5	GTC	frequent S, PS and SW (temporal and parieto-occipital region bilaterally)	Autism	6	hydromyelia (ventriculomegaly)	cough, headache, ataxia, dysphagia
4, M	6	FGTC	isolated or couple of S and SW (right central and temporal regions)	behavioral disturbance	12	Normal	no
5, F	5	A	frequent, bilateral SW and PSW	No	6	Normal	no
6, M	15	GT	left hemispheric S and SW	Autism	10	cervico-thoracic syringomyelia	no
7, M	12	GTC	fronto-central epileptiform activity with generalization	language delay	8	normal	no
8, F	6	FM	right posterior regions PSW	psychomotor delay	6	cervico-thoracic syringomyelia	central apneas
9, M	4	A	PSW	No	5	normal	no
10, M	2	A	right occipital epileptiform activity	psychomotor delay	6	hydromyelia	cough, headache
11, M	1	FM	focal right fronto-central epileptiform activity	language delay	10	normal	no
12, M	6	FNM	bilateral PSW	No	5	normal	no
13, F	7	M	slow W with S and myoclonic activity	behavioral disturbance	13	normal	cough, headache
14, F	9	GT	left temporal and posterior regions slow W and slow SW	No	6	syringomyelia (ventriculomegaly)	cough, headache, dizziness
15, M	7	FNM	right frontal SW	psychomotor delay	5	normal	cough, headache, lower limbs paresthesia
16, F	8	A	fronto-central sharp W	behavioral disturbance	5	normal	no
17, F	7	FNM	right posterior regions slow W	No	5	normal	neck pain, upper and lower limbs paresthesia
18, F	6	FM	left frontal SW	No	8.5	cervico-thoracic syringomyelia	cough, headache, neck pain
19, M	4	GT	bilateral epileptiform activity	No	6	normal	cough, headache
20, F	5	FM	right fronto-central sharp W	No	5	normal	no
21, F	6	FNM	left temporal slow W	No	6.5	normal	no
22, M	4	FM	right central SW and theta activity	No	10	normal	cough, headache, upper limbs paresthesia
23, M	8	FGTC	posterior regions biphasic S and sharp W	No	11	cervical syringomyelia	cough, headache, neck pain
24, F	7	FGTC	right frontal and temporal SW	psychomotor delay	6	normal	no
25, M	8	FM	left centro-temporal slow W and theta activity	No	5	normal	no
26, M	5	FGTC	right posterior regions slow W and S	Autism	5	normal	no
27, M	8	FNM	right centro-temporal slow W	behavioral disturbance	9	normal	no
28, F	3	FM	right central slow W	No	10	normal	no
29, F	10	FNM	left fronto-temporal S and PS	No	6	normal	no

A: absences, FM: focal motor seizures, FNM: focal non-motor seizures, FGTC: focal to generalized tonic-clonic seizures, GTC: generalized tonic-clonic seizures, GT: generalized tonic seizures, M: myoclonic seizures. PS: poly-spikes, PSW: poly-spike-waves, S: spikes, SW: spike-waves, W: waves.

**Table 2 jcm-11-06182-t002:** Characteristics of Group 2.

Patients No	Sex	Age (Years)	Signs/Symptoms	Tonsils Ectopia (mm)	Spinal Cord and Other Findings	EEG Anomalies
1	M	7	psychomotor delay, dizziness, ataxia	8	cervico-thoracic hydromyelia	theta activity
2	F	14	headache, dizziness, ataxia, dysphagia	11	normal	/
3	F	10	headache, dysphagia, dizziness	12	cervical hydromyelia	/
4	M	5	headache, upper limbs paresthesia	20	normal	/
5	F	16	cough, headache, upper limbs paresthesia	15	cervico-thoracic syryngomyelia	/
6	M	10	None	5	normal	/
7	F	15	headache	15	normal	/
8	M	6	None	14	cervical hydromyelia	/
9	F	5	psychomotor delay	13	normal	slow waves
10	M	5	psychomotor delay	11	cervico-thoracic hydromyelia	slow waves
11	F	7	psychomotor delay	13	ventriculomegaly	/
12	M	7	headache, ataxia, dysphagia	10	normal	slow waves
13	F	6	headache, neck pain	7	normal	/
14	M	3	Ataxia	10	normal	/
15	M	6	left hemiparesis	15	cervical syringomyelia	/
16	M	11	cough, headache upper and lower limbs paresthesia	10	cervico-thoracic syryngomyelia	/
17	F	3	cough, headache	7	normal	/
18	F	4	cough, headache, psychomotor delay	11	cervical hydromyelia	sharp waves, spike-waves, and poly-spikes
19	M	5	cough, headache, dizziness	11	normal	/
20	F	12	cough, headache, drop attacks	20	normal	/
21	M	7	cough, headache, psychomotor delay	9	normal	/
22	M	13	ataxia, dysphagia, dizziness, snoring	10	cervico-thoracic syryngomyelia	/
23	F	10	dyplopia, nystagmus,	12	cervico-thoracic syryngomyelia	/
24	M	4	None	5	normal	/
25	F	6	dizziness	5	normal	isolated spikes
26	M	18	hand tremors	19	cervico-thoracic syryngomyelia	/
27	F	7	cough, headache	18	normal	/
28	M	17	headache, dizziness	8	normal	/
29	F	6	cough, headache	21	normal	/
30	M	14	nystagmus, dysphagia	10	syringobulbia	/
31	M	6	dyplopia, right VI cranial nerve palsy	16	thoracic syringomyelia	/
32	F	4	ataxia, dysphagia,	12	normal	/
33	F	18	neck pain, central sleep apneas	8	normal	/
34	M	6	cough, headache	7	normal	/
35	F	8	Nystagmus	9	cervical hydromyelia	/
36	M	16	nystagmus, ataxia	20	normal	/
37	F	18	cough, headache, dizziness	15	normal	/
38	M	10	headache, dizziness	8	cervical hydromyelia	/
39	F	15	cough, headache, nystagmus	9	cervico-thoracic syryngomyelia	theta activity
40	F	4	cough, headache, dizziness	15	cervico-thoracic hydromyelia	slow waves
41	M	13	Headache	7	normal	/
42	F	7	right lower limb pain	5	normal	/
43	F	7	psychomotor delay	9	cervico-thoracic syryngomyelia	isolated spikes
44	M	5	None	7	normal	/
45	M	17	cough, headache	14	normal	/
46	M	18	headache, dizziness, lower limbs weakness	9	normal	/
47	F	10	dizziness	5	normal	sharp waves and spikes
48	F	6	dizziness	8	cervico-thoracic syryngomyelia	/
49	M	6	headache, drop attacks, vomiting	9	normal	slow waves
50	M	7	cough, headache, sialorrhoea	12	normal	/
51	F	12	Scoliosis	28	holocord syringomyelia	/
52	M	13	Headache	7	normal	/
53	F	5	headache, dizziness	11	normal	/
54	M	5	headache, dizziness	11	cervico-thoracic syryngomyelia	/
55	M	8	right hypoacusis	9	normal	/
56	F	2	cough, headache, upper limbs paresthesia	17	normal	/
57	M	3	cough, headache, central sleep apneas	5	normal	/
58	F	8	cough, headache, vomiting	8	cervico-thoracic hydromyelia	/
59	M	3	psychomotor delay	9	normal	/
60	M	15	headache, neck pain	12	ventriculomegaly	/
61	M	10	headache	10	normal	/
62	M	8	cough, headache	10	normal	/
63	F	7	headache, upper and lower limbs paresthesia	31	cervical syringomyelia	/
64	F	6	headache, enuresis	9	thoracic syringomyelia	/
65	M	14	cough, headache, lower limbs paresthesia	9	thoracic syringomyelia	/
66	F	6	Nystagmus	6	normal	/
67	M	9	cough, headache, lower limbs pain	21	cervico-thoracic syryngomyelia	/
68	M	6	headache, febrile paroxysms	12	normal	theta activity
69	F	15	headache, dizziness	18	cervico-thoracic syryngomyelia	/
70	F	7	Scoliosis	12	normal	/
71	M	16	central sleep apneas	18	normal	slow waves
72	F	5	cough, headache	13	normal	/
73	F	12	cough, headache	15	cervico-thoracic hydromyelia	/
74	M	9	headache, vomiting	8	ventriculomegaly	/
75	M	9	headache, neck pain, ataxia	7	cervical syringomyelia	/
76	M	7	ataxia, drop attacks	11	normal	/
77	M	9	cough, headache, dizziness	22	cervical syringomyelia	/

**Table 3 jcm-11-06182-t003:** Outcome of Group 1.

Patients (No)	Age (Years)	Seizures	Tonsils Ectopia (mm)	Spinal Cord and Other Findings	C1M/Syringomyelia Symptoms	Surgery	Epilepsy Outcome	C1M/Syrinx Outcome
1	7	FM	9	normal	No	none	seizure-free	Stable
2	1	GT	13	cervico-dorsal syringomyelia	cough headache, ataxia	PF decompression + tonsils coagulation	seizure-free	C1M, syrinx and headache regressed, ataxia improved
3	5	GTC	6	hydromyelia (ventriculomegaly)	cough headache, ataxia, dysphagia	ETV; bony PF decompression + C1 laminectomy	refractory seizures	C1M and symptoms regressed, hydromyelia improved
4	6	FGTC	12	normal	No	None	>90% reduction	Stable
5	5	A	6	normal	No	none	seizure-free	Stable
6	15	GT	10	cervico-dorsal syringomyelia	No	bony PF decompression + C1 laminectomy	seizure-free	C1M and syrinx stable
7	12	GTC	8	normal	No	none	seizure-free	Stable
8	6	FM	6	cervico-dorsal syringomyelia	central apneas	PF decompression + duraplasty	>90% reduction	C1M improved, syrinx and symptoms regressed
9	4	A	5	normal	No	none	>90% reduction	Stable
10	2	A	6	hydromyelia	cough headache	bony PF decompression + C1 laminectomy	>90% reduction	C1M, hydromyelia and symptoms regressed
11	1	FM	10	normal	No	none	seizure-free	Stable
12	6	FNM	5	normal	No	none	seizure-free	Stable
13	7	M	13	normal	cough headache	none	>90% reduction	Stable
14	9	GT	6	syringomyelia (ventriculomegaly)	cough headache, vertigo	ETV	seizure-free	C1M and syrinx improved, symptoms regressed
15	7	FNM	5	normal	cough headache, lower limbs paresthesia	none	>90% reduction	Stable
16	8	A	5	normal	No	none	seizure-free	Stable
17	7	FNM	5	normal	neck pain, upper and lower limbs paresthesia	none	refractory seizures	Stable
18	6	FM	8.5	cervico-dorsal syringomyelia	cough headache, neck pain	PF decompression + duraplasty	seizure-free	C1M improved, syrinx regressed, symptoms stable
19	4	GT	6	normal	cough headache	bony PF decompression + C1 laminectomy	seizure-free	C1M improved, symptoms regressed
20	5	FM	5	normal	No	none	>90% reduction	Stable
21	6	FNM	6.5	normal	No	none	seizure-free	Stable
22	4	FM	10	normal	cough headache, upper limbs paresthesia	none	seizure-free	C1M and symptoms spontaneously improved
23	8	FGTC	11	cervical syringomyelia	cough headache, neck pain	bony PF decompression + C1 laminectomy	>90% reduction	C1M and symptoms regressed, syrinx improved
24	7	FGTC	6	normal	No	none	seizure-free	Stable
25	8	FM	5	normal	No	none	seizure-free	Stable
26	5	FGTC	5	normal	No	none	>90% reduction	Stable
27	8	FNM	9	normal	No	none	seizure-free	C1M spontaneously improved
28	3	FM	10	normal	no	none	seizure-free	Stable
29	10	FNM	6	normal	no	none	seizure-free	Stable

A: absences; ETV: endoscopic third ventriculostomy; FM: focal motor seizures; FNM: focal non-motor seizures; FGTC: focal to generalized tonic-clonic seizures; GTC: generalized tonic-clonic seizures; GT: generalized tonic seizures; M: myoclonic seizures; PF: posterior fossa.

**Table 4 jcm-11-06182-t004:** Outcome of Group 2.

Patients No	Age (Years)	Tonsils Ectopia (mm)	Spinal Cord and Other Findings	EEG Anomalies	Surgery	C1M/Syrinx Outcome	EEG Anomalies Outcome
1	7	8	cervico-thoracic hydromyelia	theta activity	bony PF decompression + C1 laminectomy	C1M and syrinx improved, symptoms stable	unchanged
2	14	11	normal	/	bony PF decompression + C1 laminectomy	C1M regressed, symptoms improved	/
3	10	12	cervical hydromyelia	/	bony PF decompression + C1 laminectomy	C1M stable, syrinx regressed, symptoms improved	/
4	5	20	normal	/	bony PF decompression + C1 laminectomy	C1M stable, symptoms regressed	/
5	16	15	cervico-thoracic syryngomyelia	/	PF decompression + tonsils coagulation	C1M and syrinx regressed, symptoms stable	/
6	10	5	normal	/	none	C1M unchanged	/
7	15	15	normal	/	none	C1M unchanged	/
8	6	14	cervical hydromyelia	/	ICP monitoring	C1M and syrinx unchanged	/
9	5	13	normal	slow waves	none	C1M unchanged	unchanged
10	5	11	cervico-thoracic hydromyelia	slow waves	none	C1M and syrinx unchanged	unchanged
11	7	13	ventriculomegaly	/	bony PF decompression + C1 laminectomy	C1M unchanged, symptoms improved	/
12	7	10	normal	slow waves	bony PF decompression + C1 laminectomy	C1M and symptoms regressed	unchanged
13	6	7	normal	/	none	C1M unchanged	/
14	3	10	normal	/	none	C1M unchanged	/
15	6	15	cervical syringomyelia	/	bony PF decompression + C1 laminectomy	C1M regressed, hydromyelia improved, symptoms improved	/
16	11	10	cervico-thoracic syryngomyelia	/	PF decompression + tonsils coagulation	C1M, syrinx and symptoms regressed	/
17	3	7	normal	/	none	C1M unchanged	/
18	4	11	cervical hydromyelia	sharp waves, spike-waves, and poly-spikes	bony PF decompression + C1 laminectomy	C1M, syrinx and symptoms regressed	unchanged
19	5	11	normal	/	bony PF decompression + C1 laminectomy	C1M and symptoms stable	/
20	12	20	normal	/	PF decompression + duraplasty	C1M regressed, symptoms improved	/
21	7	9	normal	/	bony PF decompression + C1 laminectomy	C1M and symptoms improved	/
22	13	10	cervico-thoracic syryngomyelia	/	bony PF decompression + C1 laminectomy + C0–C3 fixation	C1M improved, syrinx and symptoms regressed	/
23	10	12	cervico-thoracic syryngomyelia	/	PF decompression + tonsils coagulation	C1M regressed, syrinx and symptoms improved	/
24	4	5	normal	/	none	C1M spontaneously improved	/
25	6	5	normal	isolated spikes	none	C1M unchanged	disappeared
26	18	19	cervico-thoracic syryngomyelia	/	PF decompression + tonsils coagulation	C1M and syrinx regressed, symptoms improved	/
27	7	18	normal	/	PF decompression + tonsils coagulation	C1M regressed, symptoms stable	/
28	17	8	normal	/	PF decompression + tonsils coagulation	C1M and symptoms stable	/
29	6	21	normal	/	bony PF decompression + C1 laminectomy	C1M and symptoms improved	/
30	14	10	syringobulbia	/	bony PF decompression + C1 laminectomy	C1M and syringobulbia regressed, symptoms improved	/
31	6	16	thoracic syringomyelia	/	PF decompression + tonsils coagulation	C1M, syrinx and symptoms regressed	/
32	4	12	normal	/	bony PF decompression + C1 laminectomy	C1M regressed, symptoms improved	/
33	18	8	normal	/	bony PF decompression + C1 laminectomy	C1M and symptoms regressed	/
34	6	7	normal	/	none	C1M unchanged	/
35	8	9	cervical hydromyelia	/	none	C1M unchanged	/
36	16	20	normal	/	bony PF decompression + C1 laminectomy	C1M regressed, symptoms improved	/
37	18	15	normal	/	bony PF decompression + C1 laminectomy	C1M and symptoms stable	/
38	10	8	cervical hydromyelia	/	bony PF decompression + C1 laminectomy	C1M and syrinx regressed, symptoms improved	/
39	4	9	cervico-thoracic syryngomyelia	theta activity	bony PF decompression + C1 laminectomy	C1M, syrinx and symptoms improved	unchanged
40	15	4	cervico-thoracic hydromyelia	slow waves	bony PF decompression + C1 laminectomy	C1M and syrinx regressed, symptoms improved	disappeared
41	13	7	normal	/	none	C1M spontaneously regressed	/
42	7	5	normal	/	none	C1M unchanged	/
43	7	9	cervico-thoracic syryngomyelia	isolated spikes	16 Sub + PF decompression + tonsils coagulation	C1M and symptoms worsened, syrinx stable	unchanged
44	5	7	normal	/	none	C1M stable, symptoms improved	/
45	17	14	normal	/	bony PF decompression + C1 laminectomy	C1M and symptoms improved	/
46	18	9	normal	/	bony PF decompression + C1 laminectomy	C1M and symptoms regressed	/
47	10	5	normal	sharp waves and spikes	none	C1M unchanged	unhanged
48	6	15	cervico-thoracic syryngomyelia	/	PF decompression + duraplasty	C1M improved, syrinx and symptoms regressed	/
49	6	9	normal	slow waves	bony PF decompression + C1 laminectomy	C1M improved, symptoms stable	disappeared
50	7	12	normal	/	bony PF decompression + C1 laminectomy	C1M stable, symptoms improved	/
51	12	28	holocord syringomyelia	/	PF decompression + duraplasty	C1M, syrinx and symptoms improved	/
52	13	7	normal	/	none	C1M unchanged	/
53	5	11	normal	/	ICP monitoring	C1M unchanged	/
54	5	11	cervico-thoracic syryngomyelia	/	PF decompression + duraplasty	C1M and symptoms improved, syrinx unchanged	/
55	8	9	normal	/	bony PF decompression + C1 laminectomy + C0-C3 fixation	C1M stable, symptoms regressed	/
56	2	17	normal	/	bony PF decompression + C1 laminectomy	C1M and symptoms improved	/
57	3	5	normal	/	none	C1M unchanged	/
58	8	8	cervico-thoracic hydromyelia	/	bony PF decompression + C1 laminectomy	C1M and hydrocmyelia regressed, symptoms stable	/
59	3	9	normal	/	bony PF decompression + C1 laminectomy	C1M regressed, symptoms improved	/
60	15	12	ventriculomegaly	/	ETV	C1M and ventriculomegaly improved	/
61	10	10	normal	/	bony PF decompression + C1 laminectomy	C1M and symptoms regressed	/
62	8	10	normal	/	none	C1M unchanged	/
63	7	31	cervical syringomyelia	/	PF decompression + duraplasty	C1M regressed, syrinx improved, symptoms stable	/
64	6	9	thoracic syringomyelia	/	none	C1M unchanged	/
65	14	9	thoracic syringomyelia	/	bony PF decompression + C1 laminectomy	C1M and symptoms regressed	/
66	6	6	normal	/	none	C1M spontaneously regressed	/
67	9	21	cervico-thoracic syryngomyelia	/	PF decompression + tonsils coagulation	C1M and syrinx regressed, symptoms improved	/
68	6	12	normal	theta activity	bony PF decompression + C1 laminectomy	C1M improved, symptoms stable	unchanged
69	15	18	cervico-thoracic syryngomyelia	/	PF decompression + tonsils coagulation	C1M and symptoms regressed, syrinx improved	/
70	7	12	normal	/	bony PF decompression + C1 laminectomy	C1M stable, symptoms improved	/
71	16	18	normal	slow waves	bony PF decompression + C1 laminectomy	C1M improved, symptoms regressed	disappeared
72	5	13	normal	/	bony PF decompression + C1 laminectomy	C1M improved, symptoms regressed	/
73	12	15	cervico-thoracic hydromyelia	/	bony PF decompression + C1 laminectomy	C1M and symptoms improved, syrinx regressed	/
74	9	8	ventriculomegaly	/	ETV	C1M improved and ventriculomegaly regressed	/
75	9	7	cervical syringomyelia	/	bony PF decompression + C1 laminectomy	C1M and symptoms stable	/
76	7	11	normal	/	bony PF decompression + C1 laminectomy	C1M regressed, symptoms improved	/
77	9	22	cervical syringomyelia	/	PF decompression + duraplasty	C1M and syrinx regressed, symptoms improved	/

PF: posterior fossa; ETV: endoscopic third ventriculostomy.

**Table 5 jcm-11-06182-t005:** Comparison between the two groups.

	Group 1	Group 2	*p*-Value
No patients	29	77	Not significant
Mean age at admission	6.2 years	8.9 years	0.003
M/F ratio	0.93	1.26	Not significant
Presence of seizures	All cases	None	Not calculated (patients’ selection)
EEG anomalies	All cases	17%	<0.00001
Presence of CM1/syrinx symptoms	27%	75%	Not calculated (patients’ selection)
Mean tonsillar herniation	7.5 mm	11.8 mm	<0.00001
Syringomyelia/hydromyelia	27%	39%	Not significant
Ventriculomegaly	7%	4%	Not significant
Surgery for CM1/syrinx	31%	76.5%	0.0001
Seizure-free	62%	/	Not calculated (patients’ selection)
Clinical CM1/syrinx improvement	6/9 cases	41/54 cases	Not significant
Imaging CM1 improvement	8/9 cases	41/54 cases	Not significant
Imaging syrinx improvement	7/8 cases	21/23 cases	Not significant

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
