# Peer review of "Chiari 1 Malformation and Epilepsy in Children: A Missing Relationship"

_jcm, 2022, doi:10.3390/jcm11206182_

Round 1

Reviewer 1 Report

Well written and important article. I have few comments.

1. Outcome: the term sporadic seizures is not helpful, seizure free (usually 12 months) and drug-refractory seizures are well defined terms - please amend.

2. Hydrocephalus itself may lead to seizures and epilepsy - please refer to PMID: 31563496 and discuss. Were any children treated with a shunt or only ventriculostomy.

3. Tables: the abbreviations should be better defined below the tables,.e.g. FM or ETV and all other.

Author Response

Well written and important article. I have few comments.

Answer: thank you!

  1. Outcome: the term sporadic seizures is not helpful, seizure free (usually 12 months) and drug-refractory seizures are well defined terms - please amend.

Answer: the term has been changed into "> 90% reduction".

  1. Hydrocephalus itself may lead to seizures and epilepsy - please refer to PMID: 31563496 and discuss. Were any children treated with a shunt or only ventriculostomy.

Answer: All children were treated by endoscopic third ventriculostomy. We included the digression on hydrocephalus in the discussion as far as our patients are concerned. 

  1. Tables: the abbreviations should be better defined below the tables,.e.g. FM or ETV and all other.

Answer: a legend for the abbreviations has been now included

All changes are reported in red

Reviewer 2 Report

The reviewers read the paper on Chiari malformations and epilepsy. The paper is sufficiently informative and is highly commended.

Major1.
As a final conclusion, if there is no obvious causal link between Chiari malformations and epilepsy, please consider changing the title of the article to be more clear.

Major2
I have the impression that the abstract is a little too long. Allow it if necessary, but it would be easier to understand if it were better organised as a summary.

Major3.
Epilepsy is a chronic disease resulting from electrical excitation of neurons in the cerebral cortex. Readers may wonder about the association with the Chiari malformation, which consists of cerebellar tonsils and spinal cavities; if you have an opinion on whether the Chiari malformation does or does not affect the neurons of the cerebral cortex, please add it to the discussion.

Major 4
There is a reference to the frequency of epilepsy in the Introduction. Are you sure that it is really 8%? In Asian countries, for example, it is estimated to be around 1%. Please confirm.

This is a valuable paper that contains a conclusion on the previously mentioned relationship between Chiari malformations and epilepsy.
Please make corrections in the Author's opinion and resubmit to JCM.

Author Response

The reviewers read the paper on Chiari malformations and epilepsy. The paper is sufficiently informative and is highly commended.

Answer: thank you!

Major1.

As a final conclusion, if there is no obvious causal link between Chiari malformations and epilepsy, please consider changing the title of the article to be more clear.

Answer: thanks for the suggestion. We did it.

Major2

I have the impression that the abstract is a little too long. Allow it if necessary, but it would be easier to understand if it were better organised as a summary.

Answer: actually, it is a long abstract because the article contains a lot of information. We believe that, if we maintain the abstract as it is, we can be more informative for people reading the abstract (even at a glance)

Major3.

Epilepsy is a chronic disease resulting from electrical excitation of neurons in the cerebral cortex. Readers may wonder about the association with the Chiari malformation, which consists of cerebellar tonsils and spinal cavities; if you have an opinion on whether the Chiari malformation does or does not affect the neurons of the cerebral cortex, please add it to the discussion.

Answer: we do not have evidence that Chiari can affect the activity of the brain neurons. We reaffirmed it in the manuscript. 

Major 4

There is a reference to the frequency of epilepsy in the Introduction. Are you sure that it is really 8%? In Asian countries, for example, it is estimated to be around 1%. Please confirm.

Answer. Such a high rate referred to hospitalized patients. We clarified in the manuscript and added a reference with the rate on the population (0.5.1%). 

This is a valuable paper that contains a conclusion on the previously mentioned relationship between Chiari malformations and epilepsy.

Please make corrections in the Author's opinion and resubmit to JCM.

Answer: thank you.

All changes are reported in red

Reviewer 3 Report

Massimi et al updated the information on the association between Chiari 1 Malformation (CM1) and epilepsy by reporting on a series of children specifically enrolled and retrospectively analyzed for this purpose. Despite negative results, it is very well organized study. Authors used good methodology, the paper is correctly written. However, I have several comments which can potentially improved the article. Authors should organised Tables-in this form they would fit more to the Suplementary files. Authors should grouped epidemiological and demographical data and show results in percentages and averages for each analyzed groups. What is more, I will also add poin in the discussion about potential remission of the epilepsy and factors impacting that matter (Wezyk et al. Neurol Neurochir Pol 2020;54(5):434-439.)

Author Response

Massimi et al updated the information on the association between Chiari 1 Malformation (CM1) and epilepsy by reporting on a series of children specifically enrolled and retrospectively analyzed for this purpose. Despite negative results, it is very well organized study. Authors used good methodology, the paper is correctly written. However, I have several comments which can potentially improved the article.

Answer: thank you!

Authors should organised Tables-in this form they would fit more to the Suplementary files. Authors should grouped epidemiological and demographical data and show results in percentages and averages for each analyzed groups.

Answer: we included a summary table with these characteristics

What is more, I will also add poin in the discussion about potential remission of the epilepsy and factors impacting that matter (Wezyk et al. Neurol Neurochir Pol 2020;54(5):434-439.)

Answer: we included this digression and the suggested reference in the discussion.

All changes are reported in red

Reviewer 4 Report

The authors of the article attempt to finally solve the problem of the relationship between Chiari 1 malformation (CM1) and epilepsy in children. Proper understanding of this relationship is of great importance for clinical practice, especially for prognosis of treatment progress.

The researchers compared 2 groups:

1. 29 children with CM1 who experienced seizures.

2. 77 children with CM1 with / without syringomyelia.

The authors believe that the analysis of the obtained results confirms the hypothesis of a coincidental relationship between CM1 and epilepsy. From a methodological point of view, the research was carried out correctly. This also applies to the analysis of the obtained results.

The conclusions of the authors are based on the material obtained. However, at the end of the article, there is no section on limitations related to the conducted research. Therefore, I request that the text be supplemented with this section. After making this change, I support the publication of the peer-reviewed article.

Author Response

The authors of the article attempt to finally solve the problem of the relationship between Chiari 1 malformation (CM1) and epilepsy in children. Proper understanding of this relationship is of great importance for clinical practice, especially for prognosis of treatment progress.

The researchers compared 2 groups:

  1. 29 children with CM1 who experienced seizures.
  2. 77 children with CM1 with / without syringomyelia.

The authors believe that the analysis of the obtained results confirms the hypothesis of a coincidental relationship between CM1 and epilepsy. From a methodological point of view, the research was carried out correctly. This also applies to the analysis of the obtained results.

The conclusions of the authors are based on the material obtained. However, at the end of the article, there is no section on limitations related to the conducted research. Therefore, I request that the text be supplemented with this section. After making this change, I support the publication of the peer-reviewed article.

Answer: thank you so much for the comment. A “Limitations” section has now been included in the manuscript.

All changes are reported in red

Round 2

Reviewer 1 Report

All issues resolved.

Author Response

Thank you

Reviewer 2 Report

The reviewers reviewed and read the revised paper again. The reviewer noted the revised title, the mention of relation between Chiari malformation and epilepsy in the text, and the addition of a limitation at the end of the paper, etc.

The description of the frequency of epilepsy was also separated into frequency in the general population in 1% and frequency in hospitalized cases in 8%. It's collect fact and I agree as well.

Furthermore, the correction regarding Chiari malformations and epilepsy will more clearly convey to the JCM reader, the novel message that the causal relationship between the Chiari malformations and epilepsy is not very close. And I commend the authors for following the reviewers' suggestions and making the revisions, which underscore the medical significance of the article.

There were several changes to the figures in the tables. Please provide final confirmation that these corrections are correct based on the literature. Once the confirmation is satisfactory, the reviewers will consider the paper to be an excellent paper for JCM.

Best regards,

Dr. Reviewer

Author Response

Thank you again for the comment.

We checked the changed figures: everything is ok.